# Wellbeing in Addiction Recovery: Does It Differ across Addictions?

**DOI:** 10.3390/ijerph20146375

**Published:** 2023-07-16

**Authors:** Tessa Corner, Emily Arden-Close, John McAlaney

**Affiliations:** Department of Psychology, Faculty of Science and Technology, Bournemouth University, Poole BH12 5BB, UK; tcorner@bournemouth.ac.uk (T.C.); jmcalaney@bournemouth.ac.uk (J.M.)

**Keywords:** quality of life, addiction, retrospective recall, recovery, wellbeing

## Abstract

Limited research has been conducted on the experiences of individuals in long-term recovery from addiction, and addictions are usually studied in isolation. However, no theories of addiction differentiate between addictions or assume that individuals will experience only one addiction. This study aimed to compare affect between individuals with addictions to drugs and alcohol and to explore how QoL changes in long-term recovery from addiction. Individuals in recovery from addiction (*n* = 115; 52.2% male) were recruited via snowball sampling on social media signposted by an addiction rehabilitation charity. Participants completed questionnaires about QoL (WHOQOL-Bref) and positive and negative affect (PANAS-X). The main primary addictions were drugs (76.5%) and alcohol (21.7%), with 69.7% reporting multiple addictions including food, sex, internet, and gambling. Affect and coping strategies did not differ by addiction. QoL appeared to improve with time in recovery. The high percentage of multiple addictions and greater similarities than differences between individuals with drug and alcohol addictions suggest that addictions should not be studied in isolation when studying psychological health during long-term recovery.

## 1. Introduction

Addiction to substances has been defined by the DSM-5 [1] as a psychiatric disorder leading to impaired control, physical dependence, social problems, and risky use. It is associated with depression, anxiety, thrill seeking, risk taking, incentive desensitisation, and reward deficiency [2], with recovery only occurring when the pain of continuing to use is seen as greater than the perceived pain of changing [3].

Addiction is a significant problem in the UK, affecting not only individuals but also their families and others in society involved with them. Regarding drug use, over 3000 deaths were attributed to drug misuse in 2021 [4] and male mortality from drug addiction has risen sharply since records began in 1993 [5]. Drug misuse issues are said to affect an average of 1.9 people per 100,000, with some significant variations such as Blackpool with 14.0 per 100,000 [6]. The cost of drug misuse was GBP 15.4 billion in 2014 in the UK, which included healthcare (8%), law enforcement (10%), deaths (28%), and crime (54%). It is estimated that 1.5 million people in the UK are affected by someone else’s drug addiction with the cost of harm to others estimated as GBP 1.8 billion per year. Regarding alcohol use, there were 7327 alcohol-specific deaths in the UK in 2016, making a rate of 11.7 deaths per 100,000 [6] which rose to 7565 in 2019 [7]. However, these numbers do not include any categories of death partially attributable to alcohol. This is important as alcohol impacts the innate and acquired immune system, making individuals with alcohol addiction more susceptible to infections [8]. The cost of this to the National Health Service (NHS) is around GBP 3.5 billion per annum, and there are estimated to be around 602,391 individuals who are dependent on alcohol in the UK [9].

Of those with addictions, however, many will recover and continue to live fulfilling lives. Many people live over five years in recovery (termed long-term or stable recovery). For example, 50% of the 110,095 people who left the alcohol and drug treatment system in the UK in 2020 had successfully completed treatment [10].

However, research conducted on individuals in recovery from addictions has mainly focused on participants who are on detoxification programmes, in rehabilitation programmes, or in aftercare. Limited research has been conducted on people who are no longer accessing these services but are in stable recovery. A study of 53 individuals in recovery from alcohol addictions found that those in “stable recovery”, defined as five years or more in recovery, reported better social relationships, psychological health, and environment than those in early recovery [11]. Further, individuals in stable recovery reported better social relationships and environment than population norms, supporting the idea of recovery as an ongoing gradual process with quality of life continuing to improve as abstinence duration increases. Similarly, in a nationally representative sample of US adults [12], although wellbeing increased and distress decreased most rapidly in the first five years following recovery, wellbeing continued to improve—albeit at a slower pace—for the remainder of the 40 years after entering recovery.

There are a range of theories around addiction, all of which differ widely regarding predisposition to and development of addictions. For example, biological theories see addiction as a disease of the brain. They hold that the increase in neurotransmitters caused by the intake of drugs/alcohol impairs sensitivity to natural rewards [13]; that motivation, memory, and executive function influenced by incorrect neurotransmitter regulation can reinforce learned associations, enhance the motivational value of the substance, and reduce inhibitory control [14]; and that prolonged addictive activity can cause changes in the structure and function of relevant brain regions, reinforcing addiction [15]. Predisposition theories hold that individuals may have specific vulnerabilities, whether genetic (e.g., low numbers of neurotransmitters), personality traits, stressful life experiences, or sociodemographic factors, that make them vulnerable to developing addiction [16]. Learning theories, such as classical and operant conditioning and social learning theory [17], hold that addiction is a response learned through observing others. Decision-making theories see addiction as arising from information-processing biases [18] or as a conscious choice made through cost–benefit analysis, e.g., [19,20]. Motivation theories hold that addiction may serve motives of obtaining positive rewards [21] or avoiding discomfort [22]. Psychosocial theories hold that addictions may occur to fit in with social norms [23] or to develop a social identity [24]. Self-regulation theories hold that addiction arises from deficiencies in self-control [25]. Finally, theories around contextual factors hold that vulnerability to addiction can be amplified by socio-environmental factors such as addictive behaviour among family members and peers [26,27].

Despite their differences, the theories mentioned above are concordant in not assuming that individuals will experience only one addiction [16]. Similarly, research into levels of concordant behavioural and substance addictions has identified high levels of comorbidity amongst drug users. For example, in a study of 51 participants aged 21 years or over who were currently in substance misuse treatment, over 50% reported one or more behavioural addictions concurrent to their substance addiction [28]. However, to date, research has mainly focused on single addictions with individuals with multiple addictions often being excluded, with few exceptions [29,30,31]. 

To summarise, research into experiences of addiction has several limitations. First, there is limited research on experiences of individuals in long-term recovery. Second, research tends to explore only one addiction, usually either drugs or alcohol, with individuals who have experienced multiple addictions being excluded. This study aims to compare the experiences of individuals in recovery across addictions.

We hypothesize the following:There will be no significant differences between addiction types regarding positive or negative affect.Quality of life of individuals in recovery will be similar to population norms.Quality of life of individuals in recovery will improve as length of time in recovery increases.

## 2. Materials and Methods

### 2.1. Participants and Recruitment

Snowball sampling was used to recruit participants to an online survey conducted on Qualtrics^TM^. The study was advertised on a study-specific Facebook page, signposted by a local addiction rehabilitation charity’s Facebook page and distributed via both social media and emails sent to addiction support service providers. Eligible participants were required to be aged 18 years or over, in line with ethical requirements [32], and because less than 1% of those in treatment for addiction are aged 18 years, with almost two-thirds being aged 40+ [33]. Other requirements included being currently in recovery from one or more addictions (e.g., drugs and alcohol); having access to the internet (as the questionnaire was online); and having sufficient English language ability to complete the survey. There were no other eligibility criteria. The questionnaire was anonymous as sensitive information was being collected, in line with ethical guidelines [32]. This meant that we had no means of following up with participants who reported clinically significant levels of anxiety or depression, so we considered it unethical to collect these data.

Overall, 250 participants accessed the questionnaire, of whom 162 started filling in the consent form and 154 gave consent and were eligible to participate. Of those 154 participants, 115 (74.7%) provided sufficient data for analysis. The remaining 39 stopped the study after providing demographic information. No significant differences were identified between the 115 participants who completed the questionnaire and the 39 who did not regarding age, gender, primary addiction, and length of time in recovery. Anonymised data are stored in Bournemouth University’s Online Research Data Repository (BORDAR). 

### 2.2. Measures

Participants reported their age, gender, length of time in recovery, what they considered to be their primary addiction, and any other secondary addictions they had experienced. They then completed self-report questionnaires to assess quality of life (WHOQOL-Bref) and affect (PANAS-X). 

Affect: The PANAS-X [34] consists of 60 words which assesses levels of positive and negative affect. It assesses the following emotions: general positive emotion, general negative emotion, fear, hostility, sadness, joviality, self-assurance, attentiveness, shyness, fatigue, serenity, surprise, basic positive affect (joviality, self-assurance, and attentiveness), and basic negative affect (fear, hostility, guilt, and sadness). Internal consistency (Cronbach’s alpha) is high [35]. Cronbach’s alpha in the current study ranged from 0.67 (surprise) to 0.93 (basic negative emotion) and was above 0.70, indicating very good reliability for all scales except surprise and attentiveness. Both PANAS-X and the shorter PANAS have assessed affect in individuals with substance use addictions, e.g., [36,37,38]. 

Quality of life: The World Health Organization Quality of Life Questionnaire (WHOQOL-Bref) [39] consists of 26 questions which utilise a 5-point Likert scale. Individuals rate their quality of life and satisfaction with their health. The remaining questions assess quality of life across four domains: physical health, psychological wellbeing, social relationships, and environment. It has previously been used in studies of individuals in recovery from addiction [11] and enables comparison with general population norms [40]. Reliability (Cronbach’s alpha) is good [41]. In the current study, Cronbach’s alpha was 0.19 for the physical health scale, indicating very poor reliability; 0.55 for the psychological wellbeing scale, indicating low to acceptable reliability; 0.58 for the social relationships scale, indicating low to acceptable reliability; and 0.80 for the environment subscale, indicating very good reliability. Further exploration revealed that the low reliability for the physical health scale was due to responses to the questions: “To what extent do you feel that physical pain prevents you from doing what you need to do?” and “How much do you need any medical treatment to function in your daily life” not correlating with the answers to the other questions. Further exploration revealed that removing the question “How often do you have negative feelings such as blue mood, anxiety, despair, depression?” would increase Cronbach’s alpha for the psychological wellbeing scale to 0.78, indicating very good reliability.

### 2.3. Procedure

The study was approved by the Faculty of Science and Technology Research Ethics Committee, Bournemouth University, on 30 July 2018, ref 21637. Data were collected via an online survey on Qualtrics between 8 August and 20 September 2018. 

### 2.4. Statistical Analysis

Univariate ANCOVAs were conducted to compare PANAS-X scores by addiction type, controlling for gender and length of time in recovery. Bonferroni corrections were applied to reduce the risk of type I error [42]. *t*-tests were used to compare WHOQOL-Bref scores to published norms [40]. Mixed ANOVAs and Pearson correlations were conducted to see how quality of life changed as time in recovery increased. Additionally, Bayes factors were computed in order to reduce inflation in the model. They were calculated using R (comparisons of WHOQOL-Bref scores to published norms) and SPSS (version 28, IBM Corp., Armonk, NY, USA; univariate ANCOVAs) using the comparison to a null model option and with 10,000 posterior samples. Bayes factors are interpreted in terms of the strength of support for a hypothesis. BF_01_ < 1 is regarded as strong evidence in support of the null hypothesis. BF_01_ = 1 to 3 is regarded as inconclusive evidence, BF_01_ = 3 to 10 is regarded as moderate evidence, BF_01_ = 10 to 150 is regarded as strong evidence, and BF_01_ > 150 is regarded as very strong evidence.

## 3. Results

### 3.1. Descriptives

There were 60 male participants (52.2%) and 55 female participants (47.8%). Ages ranged from 24 to 70 years (median = 48 and SD = 9.53). Time in recovery ranged from 2 months to 40 years (median = 11 years and SD = 8.34 years). Age of entering recovery ranged from 18 to 62 years (median = 34 and SD = 7.82). The median age of entering recovery was 35 years for men (SD = 6.88), 33 years for women (SD = 8.74), 36 years (SD = 9.40) for individuals with alcohol addictions, and 34 years (SD = 6.92) for individuals with drug addictions. The primary addictions were drugs (N = 88; 76.5%), alcohol (N = 25; 21.7%), food (N = 1; 0.9%), and sex (N = 1; 0.9%). Among individuals with drug addictions, there were 47 men (53.4%) and 41 women (46.6%). Among individuals with alcohol addictions, there were 13 men (52%) and 12 women (48%).

Overall, 79 (69.65%) of participants reported multiple addictions. The number of addictions per participant by gender are reported in Table 1, the number of addictions per participant by age group are reported in Table 2, and the frequency of each secondary addiction are reported in Table 3. Table 1 shows that the number of addictions did not differ by gender. From Table 3, we can see that the most common secondary addictions were alcohol and sex for men and alcohol and food for women. Significantly more women than men reported secondary food addiction (χ^2^ (1) = 8.87, *p* = 0.003). There were no other gender differences regarding the number or type of addictions. All other analyses involve comparisons between individuals with drug and alcohol addictions (*n* = 113).

### 3.2. Affect

Univariate ANCOVAs were conducted on each outcome measure with primary addiction as a between-subjects factor and length of time in recovery and gender as covariates. See Table 4 for mean scores by addiction. No main effects were identified (all ps NS). The data were then analysed using Bayesian ANOVAs. Similarly, all Bayes factors provided strong or very strong evidence favouring the null hypothesis, with BF_01_ = 53.46–925.03, indicating that affect experienced in recovery did not differ by addiction.

### 3.3. Quality of Life

The participant data were compared with normative data to aid the interpretation of the findings [40]. See Table 5. First, analyses were conducted using independent sample *t*-tests. Compared to the normative data, participants aged 30–39 years scored significantly higher in the environment domain (t (107) = −2.03, *p* = 0.045), indicating better financial resources, freedom and security, health and social care, home and physical environments, and opportunities for acquiring new skills and participation in leisure activities, whereas participants aged 50–59 years scored significantly lower in the physical domain (t (105) = 2.21, *p* = 0.03), indicating worse physical quality of life. There were no other significant differences between our participants and population norms. 

An analysis was then conducted using Bayesian statistics. For participants aged 30–39 years, the Bayes factors indicated weak evidence in support of the null hypothesis regarding the physical (BF_01_ = 1.64), psychological (BF_01_ = 2.36), and social (BF_101_ = 1.14) domains. There was weak evidence in support of the alternative hypothesis for the environment domain (BF_10_ = 1.42), indicating that our participants tended to report better quality of life than population norms. For participants aged 40–49 years, Bayes factors indicated moderate evidence in support of the null hypothesis for the physical (BF_01_ = 4.86), psychological (BF_01_ = 4.87), and environmental (BF_01_ = 4.84) domains and weak evidence in support of the null hypothesis for the social domain (BF_01_ = 2.78). For participants aged 50–59 years, the Bayes factors indicated weak evidence in support of the null hypothesis for the psychological (BF_01_ = 2.20) and social (BF_01_ = 1.85) domains and substantial evidence in support of the null hypothesis for the environment domain (BF_01_ = 4.15). There was inconclusive evidence in support of the alternative hypothesis for the physical domain (BF_10_ = 1.80), indicating that our 50–59-year-old participants tended to experience worse physical quality of life than the general population.

Participants were then split into four groups based on length of time in recovery. See Table 6. For all four domains, scores appeared to increase with recovery length up to 11–20 years and decrease during those 21+ years in recovery. This continuing increase in satisfaction scores tends to suggest that recovery is a long-term process. However, correlations between quality of life and length of recovery as a continuous variable controlling for gender were not significant (all ps NS). 

### 3.4. Differences between Individuals with One or Multiple Addictions

Univariate ANCOVAs were conducted to explore the differences in quality of life and affect between individuals experiencing one addiction and those experiencing more than one addiction, controlling for length of time in recovery and gender. There were no differences in quality of life between individuals reporting one addiction and those reporting multiple addictions. Similarly, there were no differences in affect between individuals reporting one or more addictions. Bayesian ANOVAs were also conducted. For all domains of quality of life (physical, psychological, social, and environment), Bayes factors provided very strong evidence in support of the null hypothesis, ranging from BF_01_ = 199.18 to 612.91. Similarly, for affect, Bayes factors all provided strong or very strong evidence in support of the null hypothesis, ranging from BF_01_ = 28.07 to 968.16.

## 4. Discussion

This study aimed to determine similarities and differences in relation to thoughts, feelings, and beliefs among individuals with drug and alcohol addictions. Over two-thirds of participants reported experiencing two or more addictions and the number of addictions did not differ by gender; although women were more likely than men to have a secondary food addiction. Although the most common combination was drugs and alcohol, a range of substance and non-substance-based addictions were identified. This suggests that the traditional idea of studying each addiction in isolation might not be useful [43,44,45]. In line with this idea, research has identified significantly more traumatic life events and higher anxiety levels in the lives of individuals with substance-use disorders than in those without [44]. Similarly, comorbidity between excessive internet use and problem drinking has been identified [46]. Further, male internet addicts and individuals who report alcohol dependence have similar personality traits, emotion, temperament, and increased anxiety and depression compared to healthy controls [47]. This research could be extended to assess whether similar findings apply to individuals with other addictions or multiple addictions. 

As expected, there were no significant differences between addictions regarding affect or quality of life. There were also no differences in affect or quality of life between individuals with one addiction and those with multiple addictions, even after controlling for gender. It was also expected that quality of life and health would increase over time in recovery. Participants reported high overall quality of life in recovery but only a moderate satisfaction with quality of health. Although quality of life appeared to improve as time in recovery increased until 21+ years post-recovery, when it reduced, no relation between quality of life and length of time in recovery was identified. However, it is also important to be aware that clean time increases with age. Younger participants in our study have fewer years clean, which confounds this issue. 

Compared to the general population, participants in recovery tended to report similar quality of life. There were two exceptions. First, in the domain of environment, individuals aged 30–39 years tended to score higher than population norms, possibly due to greater appreciation of life following their “second chance” [48]. Second, participants aged 50–59 years tended to score lower than population norms in the physical domain, in line with evidence that addiction accelerates the process of biological ageing [49]. However, given the low reliability of the physical health scale that suggests our participants do not have a standard physical health profile (a greater threshold for pain and belief in a reduced need for medical treatment to function relative to the general population), these results should be treated with caution.

### Limitations and Future Work

Sampling is always an issue in addiction studies. It is easier to recruit from treatment centres, aftercare services, or self-help groups [50] than from individuals who have recovered on their own, as the latter cannot be identified without self-disclosure. Having the questionnaire online only also excluded those without internet access or those who did not see the invitations sent to their treatment providers. Within the self-help community, links on social media mean that inviting one person to participate in research can lead to all of their recovery friends also seeing it through snowball sampling. These factors raise issues around the generalisability of the findings and the representativeness of the participants. Individuals who recovered naturally from addiction are likely under-represented. Although Borkman and colleagues [51] advertised in the national press to recruit individuals who had recovered from addiction naturally, 75% of their sample were from self-help groups. Further, individuals who felt their recovery was not currently successful may not have wished to participate [52]. Also, as information provided in the questionnaire was self-reported, it was not possible to exclude people who were not genuinely abstinent. Further, given the small sample size, the findings should be interpreted with caution. In addition, we did not collect data about the duration of addiction prior to entering recovery or treatment history. Quality of life in longer-term recovery may vary depending on the duration of initial addiction, given the impact of addictive substances on physical and psychological wellbeing and the evidence suggesting that individuals who receive longer-term treatment for addiction have a greater chance of sustained abstinence than those who receive shorter-term treatment [53]. Further research is needed to explore these issues. Further, we did not collect data about smoking or current alcohol intake, which is important as evidence suggests that both smoking [54] and harmful drinking [55] are significantly associated with poorer quality of life.

The finding that almost two-thirds of participants reported experiencing two or more addictions creates more questions. It is not clear whether the addictions were consecutive or concurrent. Further research is needed to determine how and why secondary addictions developed (e.g., to control primary addictions), and whether addictions progress or change over time. Coming into recovery from one addiction may also cause other addictions to become apparent. For example, the limited treatment services available for gambling addictions in the UK means that very few gamblers without concurrent addictions can access treatment. The lack of differences between individuals in recovery from drug and alcohol addictions in this study also raises questions around the importance of identifying psychological differences between groups of people who experience addiction.

## 5. Conclusions

Individuals with drug and alcohol addictions, including both men and women, report similar experiences in recovery in relation to emotions experienced and quality of life. Over two-thirds of participants considered themselves to be addicted to more than one substance or behaviour. This would suggest that, first, there are no differences between addictions regarding their impact on psychological wellbeing in the long term; second, apart from studying the effects of physical dependence, addictions should not be studied in isolation when studying psychological health during long-term recovery; and finally, quality of life appears to improve as time in recovery increases, suggesting that recovery should be thought of as a continuous process rather than a time-limited process with a clear endpoint.

## Figures and Tables

**Table 1 ijerph-20-06375-t001:** Number of addictions per participant, by gender.

Number of Addictions	Male	Female	Overall N (%)
1	18 (30%)	18 (32.7%)	36 (31.3%)
2	19 (31.7%)	13 (23.6%)	32 (21.7%)
3	7 (11.7%)	13 (23.6%)	20 (13.9%)
4	9 (15%)	7 (12.7%)	16 (13.9%)
5	6 (10%)	3 (5.5%)	9 (12.2%)
6	1 (1.7%)	1 (1.8%)	2 (6.9%)

**Table 2 ijerph-20-06375-t002:** Number of addictions per participant by age group.

Number of Addictions	Up to 29 years	30–39 Years	40–49 Years	50–59 Years	60–69 Years	70–79 Years
1	2 (50%)	7 (31.8%)	9 (23.7%)	15 (36.6%)	3 (37.5%)	0
2	1 (25%)	8 (36.4%)	11 (28.9%)	9 (22.0%)	1 (12.5%)	2 (100%)
3	0	4 (18.2%)	7 (18.45)	7 (17.1%)	2 (25%)	0
4	1 (25%)	2 (9.1%)	6 (15.8%)	6 (14.6%)	1 (12.5%)	0
5	0	0	4 (10.5%)	4 (9.8%)	1 (12.55)	0
6	0	1 (4.5%)	1 (2.6%)	0	0	0

**Table 3 ijerph-20-06375-t003:** Secondary addictions (frequency and percentage of overall sample), by gender *.

Addiction Type	Male N (%)	Female N (%)	Overall N (%)	Comparison (χ^2^)
Alcohol	28 (46.7%)	21 (38.2%)	49 (42.6%)	0.85
Drugs	8 (13.3%)	9 (16.3%)	17 (14.8%)	0.21
Food	10 (16.7%)	23 (41.8%)	33 (28.7%)	8.87 **
Sex	20 (33.3%)	11 (20%)	31 (27.0%)	2.59
Internet	11 (18.3%)	10 (18.1%)	21 (18.3%)	0
Gambling	8 (13.3%)	3 (5.5%)	11 (9.6%)	2.06
Gaming	4 (6.7%)	0	4 (3.5%)	3.80

* The percentages add up to more than 100% as some participants reported more than one addiction. ** *p* < 0.01.

**Table 4 ijerph-20-06375-t004:** Affect (PANAS-X) by addiction.

PANAS-X Variable	Primary Addiction (Drugs: n = 78) Alcohol: n = 22)	Score (M, SD)
General Positive	Drugs	3.16 (1.12)
	Alcohol	3.01 (0.74)
General negative	Drugs	1.66 (0.61)
	Alcohol	1.66 (0.58)
Fear	Drugs	1.54 (0.60)
	Alcohol	1.67 (0.67)
Hostility	Drugs	1.86 (0.49)
	Alcohol	1.86 (0.40)
Guilt	Drugs	1.60 (0.76)
	Alcohol	1.58 (0.81)
Sadness	Drugs	1.78 (0.80)
	Alcohol	1.55 (0.52)
Joviality	Drugs	3.54 (0.77)
	Alcohol	3.54 (0.45)
Self-Assurance	Drugs	3.48 (0.63)
	Alcohol	3.33 (0.63)
Attentive	Drugs	3.24 (0.50)
	Alcohol	3.23 (0.51)
Shyness	Drugs	1.70 (0.65)
	Alcohol	1.84 (0.59)
Fatigue	Drugs	2.08 (0.79)
	Alcohol	2.13 (0.68)
Serenity	Drugs	3.61 (0.76)
	Alcohol	4.00 (0.68)
Surprise	Drugs	2.71 (0.82)
	Alcohol	2.61 (0.88)
Basic positive	Drugs	3.58 (0.64)
	Alcohol	3.59 (0.45)
Basic negative	Drugs	3.16 (1.12)
	Alcoholics	3.01 (0.74)

**Table 5 ijerph-20-06375-t005:** Quality of life (WWHOQOL-Bref) relative to population norms.

Age	Group	N	Physical (Mean, 95% CI)	Psychological (Mean, 95% CI)	Social (Mean, 95% CI)	Environmental (Mean, 95% CI)
30–39	Normative data	87	82.0 (79.1–84.9)	73.5 (70.5–76.5)	73.7 (69.6–77.8)	73.2 (70.5–75.9)
	Participants	22	76.7 (70.9–87.2)	69.7 (63.1–76.2)	65.9 (58.0–73.9)	79.3 (73.1–85.5) *
40–49	Normative data	88	77.8 (73.6–82.0)	71.5 (68.4–74.6)	72.1 (68.3–75.9)	72.3 (69.6–75.0)
	Participants	38	78.1 (73.8–86.9)	71.4 (66.3–76.3)	68.2 (62.5–74.5)	76.4 (71.8–81.1)
50–59	Normative data	66	80.3 (76.1–84.6)	73.8 (70.7–76.9)	73.1 (68.6–77.6)	77.0 (73.7–80.3)
	Participants	41	72.6 (68.1–79.5) *	70.0 (64.8–74.7)	67.5 (62.5–74.5)	78.5 (73.7–82.9)

* Participants’ results differ significantly from population norms.

**Table 6 ijerph-20-06375-t006:** Quality of life (WWHOQOL-Bref) in recovery by length of recovery for individuals with alcohol and drug addictions.

Recovery (Years)	N (% by Gender)	Addiction	Physical (Mean, 95% CI)	Psychological (Mean, 95% CI)	Social (Mean, 95% CI)	Environmental (Mean, 95% CI)
0–5	33 (29.2%) M: 15 (13.3) F: 18 (15.9)	Drugs: 20 (17.7%) Alcohol: 13 (11.5%)	72.4 (65.5–79.3)	66.8 (60.9–72.7)	65.5 (58.3–72.7)	74.6 (69.1–80.2)
6–10	23 (20.4%) M: 12 (10.6) F = 11 (9.7)	Drugs 19 (16.8%) Alcohol 4 (3.5%)	74.5 (66.5–82.4)	70.0 (63.2–76.8)	62.7 (54.4–71.0)	77.2 (70.8–83.6)
11–20	39 (34.5%) M: 19 (16.8) F: 20 (17.7%)	Drugs 34 (30.1%) Alcohol 5 (4.4%)	75.4 (69.3–81.6)	70.6 (65.3–75.9)	69.3 (62.9–75.8)	78.8 (73.8–83.8)
21+	18 (15.9%) M: 13 (11.5) F: 5 (4.4)	Drugs 15 (13.3%) Alcohol 3 (2.7%)	74.1 (63.8–84.4)	66.6 (57.8–75.3)	68.9 (58.3–79.6)	74.3 (66.1–82.6)

## Data Availability

The data presented in this study are openly available in Bournemouth University’s online data repository, BORDAR, at https://doi.org/10.18746/bmth.data.00000271 (accessed on 29 November 2022).

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
