# Peer review of "Wellbeing in Addiction Recovery: Does It Differ across Addictions?"

_ijerph, 2023, doi:10.3390/ijerph20146375_

Round 1
Reviewer 1 Report
The paper has a problem in the size of the sample related to the methodology to obtain it, from this point of view the results must be analyzed with prudence, and this problem must be point up in the limitations of the paper. The introduction and methodology looks corrects. The bibliographic revision is adequate
Author Response
The paper has a problem in the size of the sample related to the methodology to obtain it, from this point of view the results must be analysed with prudence, and this problem must be point up in the limitations of the paper. The introduction and methodology look correct. The bibliographic revision is adequate.
Thank you for your comments. We are aware the results are based on a small sample. We have added discussion of the small sample size to the limitations section, where we state “Further, given the small sample size, the findings should be interpreted with caution.”
Reviewer 2 Report
This study compared emotional symptoms and quality of life in those with long term recovery from drugs and from alcohol, also compared those with one addiction and those with multiple, and found no group differences. This study focus on long-term recovery which is an important topic, however, study has some main issues that need to be addressed.
1. In the Participants and Recruitment section, only 115 out of 250 provided sufficient data. This means only less than 50% data were used for the study. Authors need to describe in detail what data were missing for the 115 and whether those 115 had different demographics than those who provided completed data.
2. For the descriptive data analysis, author should report median instead of mean for age, age at recovery, length of recovery, etc., due to the large variability on these variables in the current study.
3. Some information are important but missing, authors should provide sex proportion and age group proportion for alcohol/drug addiction and for single/multiple addiction. Also, when separating participants by length of recovery, how many male/female, alcohol/drugs in each category?
4. A main limitation of statistic methods is not include sex as an covariate. Female is known to suffer more than male from addiction. Female with addiction usually have more emotional distress, severe addictive symptoms, and less successful treatments compared to male.
5. Another important factors that affect recovery includes severity of addiction. Although authors stated that the participants were recover at home, but they didn't clarify all questions that were asked during the survey. For example, duration of addiction, treatment history, any social problems caused by their addiction, such as at work or family. All these issues directly affect one's emotional health and quality of life.
6. Authors should explain their finding more clear. For example, participants had higher than norm scores in the Environment domain, what does this finding mean?
7. For this conclusion "scores appeared to increase with recovery length up to 11-20 years 184 and decrease for those 21+ years in recovery.", did authors at least try to run some correlation between length of recovery and the outcome measures? Again, this correlation also needs to take account sex difference.
8. In the conclusion, it is not clear what findings form this conclusion "rather than there being many types of addictions, 260 “addiction” presents in many variations." given than the hypotheses and findings all pointed to "no difference" between addiction types.
9. Again in the conclusion, the sentence "apart from studying the effects of physical dependence, addictions should not be studied in isolation" is also very vague. Authors should define this point further, such as "...in isolation when study psychological health during long-term recovery" or something similar and more consistent with the findings of the current study.
Language is fine, but missing some punctuation.
Reviewer 3 Report
Thank you for the opportunity to review this study. This is a very timely and intriguing work and I read with enthusiasm. Although there are no several concerns, I’d highly suggest a few things.
1) Double check your references accordingly to the Journal’s standards;
2) I missed some background on smoking and high-order processes that affect quality of life per se. For example, background on smoking / nicotine intake on attention etc. and other functions;
3) Please provide the Cronbach values;
4) Also I think the authors need to extend their eligibility criteria (more details are needed);
5) The stats section seems not appropriate. Although I understand the choices, the Bayesian is really more interesting for your data. The same for the linear mixed model using Bayesian, which is fine and quick using R or other software;
6) Finally, conclusions need to be refined and the strengths and limitations should be placed with further information for other researchers and readers;
7) Please work on conciseness but also readability throughout the file
Round 2
Reviewer 2 Report
The manuscript improved significantly after revision, I do not have more comments.
Author Response
We thank the reviewer for their comments.
Reviewer 3 Report
I would to thank the authors for their edits.
Nevertheless, it’s a surprise the authors did not get the point on “how to extend eligibility” or even why to change the analyses. Also, how attention and other higher order processes can affect quality of life? Just a quick search and the authors will find - the relation with intake is very highlighted in those studies.
1: How extend eligibility? The sample was controlled for confounding factors? Which? For covid, infection, mental state, use of other substances etc? Then place the references. The authors mention 18 years, so it’s clear that’s a ethic’s consensus - the same is applied to the others;
2. Bayesian is a robust and used stats used recently because of a number of benefits, one of them would be remove the inflation of your model;
3. Always remember that a rebuttal should be followed with arguments not with “we are not sure why or how… then we’d stick with our old statements”.
Since basically nothing was refined at all, neither readability and conciseness, I am sorry I can’t be of many help now.
Author Response
Response to Reviewer 3
I would to thank the authors for their edits. Nevertheless, it’s a surprise the authors did not get the point on “how to extend eligibility” or even why to change the analyses. Also, how attention and other higher order processes can affect quality of life? Just a quick search and the authors will find - the relation with intake is very highlighted in those studies.
Many thanks for your comment. Apologies, but we did not fully understand what the reviewer meant by the point around extending eligibility. Regarding attention and higher order processes, we agree with the reviewer that these factors can impact quality of life. However, there are a number of factors that can impact quality of life, and it was necessary to narrow down the topics to focus on, to avoid burdening our participants.
1: How extend eligibility? The sample was controlled for confounding factors? Which? For covid, infection, mental state, use of other substances etc? Then place the references. The authors mention 18 years, so it’s clear that’s a ethic’s consensus - the same is applied to the others;
Thank you for clarifying what you meant by extending eligibility. We did not control for COVID-19 infection as the study was conducted in 2018, and the first case of COVID-19 was identified in 2019. Information about study dates is stated in the Procedure section.
We recruited participants aged 18 years or over for two main reasons. The first was ethical: in the UK individuals are considered adults at age 18, and able to give independent consent. Second, the study aimed to recruit individuals in recovery from addiction and very few individuals in recovery are aged under 18 years. We have added this information to the “Participants and Recruitment” section, where we state:
“Eligible participants were required to: be aged 18 years or over in line with ethical requirements (British Psychological Society, 2021) and because less than 1% of adults in treatment for addiction are aged 18 years, with almost two thirds being aged 40+ (Office for Health Improvement and Disparities, 2023); currently be in recovery from one or more addictions (e.g., drugs, alcohol); have access to the internet (as the questionnaire was online); and have sufficient English language ability to complete the survey.”
We did not collect data on mental state for ethical reasons – because the questionnaire was anonymous, we had no means of following up participants who reported clinically significant levels of anxiety or depression. We have added information about this to the “Participants and Recruitment” section, where we state:
“The questionnaire was anonymous as sensitive information was being collected, in line with ethical guidelines (British Psychological Society, 2021). This meant we had no means of following up participants who reported clinically significant levels of anxiety or depression, so considered it unethical to collect this data.”
Regarding use of other substances, we collected data on what substances the participants considered themselves to be in recovery from, including their primary addiction and any other addictions. However, we did not collect data about smoking and alcohol. We agree this is a limitation and have added this point to the Discussion section, where we state:
“Further, we did not collect data about smoking or current alcohol intake, which is important as evidence suggests that both smoking [54] and harmful drinking [55] are significantly associated with poorer quality of life.”
- Bayesian is a robust and used stats used recently because of a number of benefits, one of them would be remove the inflation of your model;
Many thanks for your comment. We agree regarding the strengths of Bayesian statistics. Unfortunately, we were given limited time to complete the revisions. We have now rerun the analyses using Bayesian statistics.
- Always remember that a rebuttal should be followed with arguments not with “we are not sure why or how… then we’d stick with our old statements”.
We thank you for your comment. Apologies, but some of the “we are not sure” comments were because we did not fully understand the meaning of the comments. If we had clearly understood what was required, we would have devoted more time and effort to answering the comments. Thank you for clarifying your comments in this round of revisions.
